



# Using paired teaching for earthquake education in schools

Solmaz Mohadjer[1,4], Sebastian G. Mutz[1], Matthew Kemp[2], Sophie J. Gill[2], Anatoly Ischuk[3], Todd A. Ehlers[1]

[1]Department of Geosciences, University of Tübingen, Tübingen, 72074, Germany
[2]Department of Earth Sciences, University of Oxford, United Kingdom
[3]Institute of Geology, Earthquake Engineering and Seismology, Dushanbe, Tajikistan
[4]Earth and Environmental Sciences, University of Central Asia, Khorog, Tajikistan

*Correspondence to*: Solmaz Mohadjer (solmaz.mohadjer@uni-tuebingen.de)

**Abstract.**

Lack of access to science-based natural hazards information impedes the effectiveness of school-based disaster risk reduction education. To address this challenge, we have created ten geoscience video lessons that follow the paired teaching pedagogical approach. This method is used to supplement the standard school curriculum with video lessons instructed by geoscientists from around the world coupled with activities carried out by local classroom teachers. The video lessons introduce students to the scientific concepts behind earthquakes (e.g., Earth's interior, plate tectonics, faulting, and seismic energy), earthquake hazards and

mitigation measures (e.g., liquefaction, structural and non-structural earthquake hazards). These concepts are taught through hands-on learning where students use everyday materials to build models to visualize basic Earth processes that produce earthquakes, and explore the effects of different hazards. To evaluate the effectiveness of these virtual lessons, we tested our videos with school classrooms in Dushanbe (Tajikistan) and London (United Kingdom). Before and after video implementations, students completed questionnaires that probed their knowledge on topics covered by each video including the Earth's interior, tectonic plate

boundaries, and non-structural hazards.

Our assessment results indicate that while the paired teaching videos appear to enhance student views and understanding of some concepts (e.g., Earth's interior, earthquake location forecasting, and non-structural hazards), they bring little change to their views on causes of earthquakes and their relation to plate boundaries. In general, the difference between UK and Tajik students' level of knowledge prior to and after video testing is more significant than the difference between pre- and post-knowledge for each group.

This could be due to several factors affecting curriculum testing (e.g., level of teachers' participation and suitable classroom culture) and students' learning of content (e.g., pre-existing hazards knowledge and experience). Taken together, to maximize the impact of school-based risk reduction education, curriculum developers must move beyond innovative content and pedagogical approaches, take classroom culture into consideration, and instil skills needed for participatory learning and discovery.

## 1 Introduction

The impacts of earthquake disasters are not only physical, psychological and economic, but also educational. Approximately 1.2 billion students are enrolled in primary and secondary schools, with about 875 million living in high seismic zones (UNICEF, 2014). Recent devastating earthquakes such as those that struck Pakistan in 2005, China in 2008, Haiti in 2010 and Nepal in 2015 have demonstrated how vulnerable school communities are to earthquake disasters (Fig. 1). In the most affected regions, these earthquakes resulted in the collapse of over 80% of schools (Pazzi et al., 2016). In China alone, the Wenchuan earthquake destroyed

more than 7,000 school buildings and significantly damaged more than 10,000. The number of school children affected was estimated to be in the millions. Similarly, in Pakistan, UNICEF reported at least 17,000 school children were killed, most of them





in the collapse of more than 7,500 school buildings, as well as some 2000 teachers lost their lives, were seriously injured or displaced (Winser, 2005; Halvorson and Hamilton, 2010).

Scarce resources, inadequate building codes, and unskilled building professionals are often cited as the underlying drivers for
unsafe school buildings (Sharma et al., 2016; Bilham and Gaur, 2013; Erdik and Durukal, 2008). Other contributing factors include a lack of science-based earthquake education, awareness of hazards and mitigation measures, and sociocultural factors influencing knowledge, beliefs and practices (Lownsbery and Flick, 2020; Cavlazoglu and Stuessy, 2013; Halvorson et al., 2007). In some societies, the lack of access to science-based earthquake information can hinder preparedness by cultivating misconceptions such as those relating to fatalism and God's will (Yeri et al., 2019; Paradise, 2005) or by blaming and shaming of the vulnerable
population (Simpson, 2011; Halvorson and Hamilton, 2007). Previous work has identified school based disaster risk reduction (DRR) education as one of the main contributors to the long-term resilience and empowerment communities (Subedi et al., 2020; Oktari et al., 2018; UNICEF, 2014; Twigg, 2009). An effective DRR curriculum can prepare children and youth as "agents of change" by actively engaging them in learning about geo-hazard science and school safety measures, and preparing them to share their learning with the wider community (see Mitchell et al., 2009, and references within). However, many schools around the
world lack the resources and incentives that are required for a successful school-based DRR education, for example curricular resources, and trained teachers. This study aims to improve teachers' access to DRR content focusing on earthquake education, and to facilitate content teaching by connecting teachers with Earth scientists.

Different approaches to school-based DRR curriculum are summarized in a comprehensive report by UNICEF/UNESCO which includes case studies from thirty countries (Selby and Kagawa, 2012). In general, curriculum development and integration is
textbook-driven and/or carried out as co- or extra-curricular activities in the form of pilot projects or special events in schools (e.g., earthquake drills). According to the above report, some DRR content is easily woven into specific school subjects such as geography or natural sciences. However, a textbook-driven approach hinders the achievement of skills, attitudinal and action learning outcomes required for effective DRR learning. Co-curricular and extra-curricular activities, on the other hand, can provide a means for innovative teaching and interactive and participatory learning. Here, we apply an innovative teaching technique, known
as paired teaching and/or teaching duet, to enhance science-based earthquake education that also covers topics related to earthquake hazards and safety measures. The curriculum is designed to supplement the standard school curriculum with virtual lessons instructed by Earth scientists and hands-on activities carried out by in-class teachers.

The curriculum content (in the form of lesson plans) was originally developed and tested in school classrooms in Tajikistan (Mohadjer et al., 2010) and used by Teachers Without Borders and other educational institutions in the training and guidance of
teachers to facilitate DRR learning in schools in China, Haiti, Afghanistan, Tajikistan and India. Here we present how we applied the teaching-duet technique to the curriculum and share results of our curriculum classroom testing in schools in Tajikistan and the United Kingdom.

## 2 Methods

We created and published ten free (Attribution-NonCommercial Creative Commons license) online video lessons. The videos are
archived with Technische Informations Bibliothek (TIB) AV-Portal (https://av.tib.eu/media/47600) and can be accessed via the YouTube channel of the European Geosciences Union (https://www.youtube.com/user/EuroGeosciencesUnion). The video series was created collaboratively by nine early-career Earth and environmental scientists from academic institutions across the United



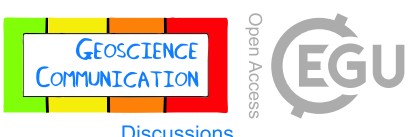

Kingdom and Germany. To create the series, we adapted protocols for creating interactive videos by Larson and Murray (2017) and applied them to lessons plans developed by Mohadjer et al. (2010). These protocols incorporate the paired teaching pedagogy

(see section 2.1), and blend well with the interactive exercises that accompany the lesson plans. For a through discussion on the protocols, we refer readers to Larson and Murray (2017). Below, we introduce the paired teaching approach, provide a video example, and describe our video evaluation strategy.

## 2.1 Paired teaching pedagogy

We used a pedagogical model known as paired teaching (or teaching-duet) developed by the MIT BLOSSOMS (Blended Learning

Open Source Science or Math Studies) initiative. This approach enables scientists and educators from around the world to create and instruct virtual lessons and activities that are carried out under the guidance of in-class teachers in school classrooms. A typical virtual lesson contains four to six short video segments taught by the video-teacher. Each segment is followed by a live active-learning segment in the classroom, guided by the in-class teacher. For example, the class starts with segment 1 of a learning video. Toward the end of this segment, the video-teacher gives a challenge to the class. The video fades into black and is replaced with a

still image of activity instructions or questions. The in-class teacher pauses the video, and guides the students in an active learning exercise in the classroom. After the exercise is concluded, the in-class teacher resumes the video, allowing the video-teacher to take over the teaching. The passing of teaching between the in-class and video-teachers is an iterative process and a type of blended learning referred to as the teaching duet or paired. Fig. 2 shows the workflow of our paired teaching approach.

In this study, the above technique was adapted and applied to earthquake education lesson plans of Mohadjer et al. (2010). To turn these lesson plans into learning videos, we followed protocols created by MIT BLOSSOMS. The protocols included several steps before videotaping began: the writing of the concept for the video lesson, mapping out the structure and content of the video lesson, and preparing a pseudo-script to show how the video lesson will be presented when taped. The lesson plans and learning videos cover similar topics, but they differ in both teaching pedagogy and the delivery method. In contrast to our paired-teaching videos,

the lesson plans followed the 5E (Engagement, Exploration, Explanation, Elaboration, and Evaluation) Instructional Model of Bybee et al. (2006) and were produced as printable instructions for teachers.

### 2.1.1 Paired teaching video example

To illustrate the paired teaching approach used in this study, we provide below an example of a video module from our earthquake video series: *Journey to the Centre of the Earth* by Matthew Kemp from the University of Cambridge. The goal of this video lesson

is to introduce students to fundamental scientific concepts behind earthquakes (i.e., plate tectonics and the driving mechanisms) in order to prepare them for topics covered in later videos (e.g., earthquake anatomy, associated hazards and mitigation measures).

The video module contains five video segments with five active learning exercises that take place in the classroom under the guidance of the in-class teacher. The active learning segments are based on questions and/or activities proposed by the video-

teacher at the end of each video segment. In addition, a teacher's guide segment is included at the end of the video to help the in-class teacher prepare for the lesson. This video module is designed for a 60-minute-long class, with approximately 22 minutes given to the video-teacher and 28 minutes given to the in-class teacher for group exercises. Each video segment takes between 2-5 minutes. The in-class teacher (1) starts playing the video with students sitting in the classroom, (2) pauses the video at the end of each video segment, (3) facilitates the active-learning session with students in the classroom, and (4) resumes the video when

the active-learning exercise is completed. The details of each segment follows.



*Segment 1 (2 minutes)*: The video-teacher introduces himself. He then holds on to a large inflatable globe and shares some intriguing numbers about the size of the Earth (e.g., it would take about 11 years to walk around the Earth) and the Earth's spinning speed around its own axis and the Sun. He then entices students into thinking about the Earth's interior by holding a wrapped

present and telling them, "[…] soon it is going to be my birthday and someone has kindly bought me a present, and of course I want to find out what is inside the present before that day itself." He tries out different strategies to figure out the content of the present (e.g., shaking it and holding it up to the light). He then pauses and asks students, "but what if I told you that I could never open this present? How would I actually find out what the present was?" Then he points at the inflatable globe, "that's a bit like how it is for scientists and the Earth. The Earth is like a gigantic present that they're never going to unwrap." He continues, "despite

this difficulty, scientists know a lot about the Earth's interior structure and composition, and one way to study the Earth's interior is to drill a deep hole." The video-teacher ends the segment by asking students to discuss the following question in groups and under the guidance of their in-class teacher:  How deep have scientists drilled into the Earth?  [The video fades into black, the in-class teacher pauses the video and takes over the teaching]

*Segment 2 (2 minutes)*: The video-teacher gives some insights into the question asked in segment 1, including sharing examples of drill projects from around the world and explaining some of the factors that hinder the drilling depth. To give a sense of scale, the video-teacher compares the deepest hole drilled into the Earth to biting an apple and barely breaking through its skin. The teacher continues keeping students interested by stating that despite not being able to drill deep into the Earth's interior, scientists know a lot about its internal structure and composition. At the end of this segment, the teacher asks students to discuss the following

question: What other methods do scientists use to learn about the interior of the Earth? [The video fades into black, the in-class teacher pauses the video and takes over the teaching]

*Segment 3 (2 minutes):* The video-teacher recalls how in segment 1 he tried to find out about the content of the wrapped present by shaking it and listening to the sounds he could hear. The video teacher relates this to how scientists use seismic waves to map

the interior structure of the Earth. At the end of this segment, the video-teacher asks students to learn about different types of seismic waves and use a Slinky to model each. [The video fades into black, the in-class teacher pauses the video and takes over the teaching]

*Segment 4 (5 minutes):* In this segment, the video-teacher uses a 3D model of the Earth's cross section as a visualization tool to

point out and describe each layer, and what (and how) scientists know about its composition. The video-teacher uses the Slinky to remind and demonstrate how seismic waves behave as they travel through different materials, and how scientists use that information to map each layer. The video-teacher ends this segment by inviting students to use a hard-boiled egg as a model of the Earth to list the different layers of the egg and relate these layers to those of the Earth. [The video fades into black, the in-class teacher pauses the video and takes over the teaching]


*Segment 5 (4 minutes):* The video-teacher shows a hard-boiled egg cut in half and relates each egg layer to a layer in the Earth. He then gently cracks the egg to simulate plate tectonics and plate movements. He points out the limitations associated with the egg model, and summarizes the lesson content. At the end of this segment, the video-teacher invites students to discuss the limitations of the egg model. He proposes a series of questions to guide the discussion: (1) how to modify the egg model to overcome its

weaknesses, (2) what are the strengths of the egg model, (3) can you think of better analogies for the Earth, and (4) what did the egg model teach you that you didn't know already? [End of video intended for classroom use]





*Teacher Segment (5 minutes):* The goal of this segment is to assist the in-class teacher in directing the learning that takes place in the classroom. It is intended that this teacher segment would be viewed by the teacher prior to using the rest of the video for paired

teaching. The segment starts with the video-teacher stating the lessons' pre-requisites and materials needed for classroom activities, and encouraging the in-class teacher to contextualize the content by incorporating examples and analogies that are more appropriate and relevant to students' lives. In addition, background information about seismic waves and how to produce them using a Slinky or an alternative teaching demo are shared.

**2.2 Curriculum evaluation**

To evaluate the effectiveness of our video series, we compared pre- and post-assessment data collected on the first and last days of video implementation, respectively. We did this using a questionnaire that assessed students' learning of main concepts covered in each video that was selected for classroom implementation. This comparison is possible since students answered the same questions in pre- and post-assessment questionnaires. Below, we describe the questionnaire, our video selection, and testing sites.

**2.2.1 Pre- and post-assessment questionnaires**

The questionnaire (see supplementary material) contained seven questions, with each question designed to evaluate students' learning of a topic that was explored in selected videos. To elicit a wide range of responses from the students, we used open-ended questions (e.g., what are the causes of earthquakes?), drawing strategies (e.g., draw the Earth's interior) and analysis of photographs (e.g., can you identify non-structural hazards in each photo?). We also included questions requiring a "yes" or "no" answers (e.g.,

Can earthquakes be predicted?). For these questions, we included "I don't know" and "other (please specify)" in answers to select from. The only demographic information collected was student gender. To anonymously link the pre- with post-assessment data collected from each individual student, we asked students to create a confidential identifier unique to themselves, and write it on both pre- and post-assessment questionnaires. Using this method, we were able to collect data from a total of 77 students from Tajikistan and the UK.

**2.2.2 Data Analysis**

Students' written responses to survey questionnaires were first categorized into appropriate groups based on the individual response to each question. We then assigned a score value to each response group, indicating the level of understanding associated with the response. For examples, students who mentioned volcanoes and mountains in their answers as the primary cause of earthquakes received a score of 1; those mentioning plate tectonics received a score of 2; those with no or irrelevant answers received a score

of 0. Similarly, students' graphical responses (question 6 in the survey) were analysed using the evaluation rubric of Steer et al. (2005). For example, students' drawings of the Earth's interior were scored 0 (no conceptual framework) to 5 (advanced understanding). Students' scored responses to pre- and post-assessment questions were then analysed for comparison using a two-sample t-test when data followed the normal probability distribution and the non-parametric Kolmogorov-Smirnov (KS) test when the probability distributions were non-normal. For survey questions that required binary answers (yes/no questions such as

questions 3-4 in the survey), we used the McNemar test to compare results. The significance level (alpha value) was set to 0.05, and results were considered statistically significant if $p < 0.05$.





### 2.2.3 Video selection

We selected three videos for classroom testing in the UK and Tajikistan. These videos included the first two earthquake science

video lessons in the series (i.e., Earth's interior and Plate boundaries) and the last earthquake hazard and safety video (i.e., Non-structural hazards) (see Fig. 3). These videos were chosen since they required no previous knowledge of earthquakes, and cover the fundamental concepts related to earthquakes (i.e. Earth's interior, plate tectonics) and the most common cause of earthquake-related injuries (i.e., non-structural hazards). Furthermore, these videos use a wide range of pedagogical approaches to teaching and learning. For example, the Earth's interior video follows a model-based, conceptual change approach to teaching to improve

students' understanding of Earth's structure while the plate boundaries video is data-driven and follows the jigsaw method of cooperative learning (i.e., students depend on each other to succeed) as shown in Sawyer et al. (2005). The non-structural hazards video uses a place-based approach, promoting learning that is rooted in students own "place" (i.e. their classroom) to raise awareness about non-structural hazards and mitigation measures in school classrooms where videos were tested. This range of teaching methods tested allows us to gain the most information about students' diverse and complex individual learning needs in

response to paired teaching, from the information collected in the questionnaires.

### 2.2.4 School settings

The three aforementioned videos were tested with 38 sixth-grade students (12 years of age, 50% female) and 39 ninth-grade students (12-14 years of age, 42% female) from two school classes in Dushanbe (Tajikistan) and London (United Kingdom), respectively. The school in Dushanbe (capital city of Tajikistan) is a public school located in the city center and was selected for

this study by the Tajik Institute of Earthquake Engineering and Seismology because of its previous collaboration with Mohadjer et al. (2010). The school was recently renovated. Included in the renovations were the installation of interactive whiteboards (IWB) in many of its classrooms, and a new computer lab with individual workstations. The video testing was conducted over 5 days during school hours (i.e. 8:00-12:00) by the lead author with assistance from local teachers. The local language (Tajik) was used in teaching and in all written and media materials. Resources needed for testing (e.g., maps, Slinky toys and Playdough) were

provided by the lead author. The school in London (United Kingdom) is a secondary (11-18 years of age) academy located in the heart of the city. The London school was selected through our existing teachers' network in the UK. The videos were tested by two geography teachers with 1-3 years of teaching experience over a testing period of approximately 50 days.

### 3 Results

Using the methods described in section 3, we created ten online video lessons. The videos were tested with school classrooms in

Tajikistan and the United Kingdom during the 2018-2019 period. Below, we briefly introduce the videos and share the results of our classroom implementation.

### 3.1 Paired-teaching video series

Table 1 summarizes the teaching approach, topics covered, and classroom activities for each video as well as video duration and its digital object identifier (doi). The video durations range from 12 to 24 minutes (excluding the teacher segment). In addition to

our paired-teaching pedagogy, each video lesson incorporates a wide range of teaching strategies to create an active learning environment. While some strategies are based on group work methods such as cooperative learning (as used in the *Discovering Plate Boundaries* video lesson), others include classroom experiments that involve students in collecting data, making predictions and reflecting upon their observations (as shown in the *Earthquake Machine* video lesson). Most video lessons incorporate





analogies and models to enhance conceptual understanding of some topics such as Earth's interior structure and material properties

while allowing students to construct and critique their models. All classroom activities are low-tech and require materials that can be easily obtained and assembled anywhere in the world.

### 3.2 Video classroom testing

To test the effectiveness of our paired teaching technique, we selected and tested three video lessons with school students in Tajikistan and the United Kingdom. Since both groups watched the same videos and completed the same questionnaires prior to

and after the testing of video lessons, full comparison of results between groups is possible. In the following sub-sections, we summarize students' responses to six questions they were asked in the pre- and post-assessment surveys.

### 3.2.1 Students' understanding Earth's interior

Students' understanding of the Earth's interior before and after classroom testing of the *Earth's Interior and Plate Tectonics* video is shown in Fig. 4. While the majority of the Tajik students' responses, both before and after watching the video, show no to little

understanding of the Earth's interior (scores of 0-2), responses given by the UK students are more evenly distributed between no/little understanding and higher levels of understanding (scores of >2). Both groups, however, show an increase in their understanding of the Earth's interior after watching the video. These observations are supported by statistical analysis of the results. More specifically, a notable percentage of students from Tajikistan (74%) and from the UK (48%) demonstrated having a no/naïve conceptual framework about the Earth's interior (scored 0-1) prior to video testing. After video testing, a large percentage of Tajik

and UK students (58% and 52%, respectively) demonstrated an increased level of understanding of the Earth's interior (scored 3 or higher). The difference between Tajik students' responses before and after video testing was statistically significant above 95% level using the Kolmogorov-Smirnov (KS) test (D-stat: 0.31, D-crit: 0.30). Similarly, the difference between UK and Tajik students' responses before and after video testing was significant above 95% (D-stat: 0.33 and D-crit: 0.30).

### 3.2.2 Students' understanding of causes of earthquakes

We grouped students' responses into six categories (Fig 5). In their responses to "What are the causes of earthquakes?", 90% of UK students mentioned plate tectonics while 46% of Tajik students made references to mountains and volcanoes (with only 2% mentioning plate tectonics) before video testing. After video testing, Tajik students showed little improvement in their understanding of the causes of earthquakes. The difference between UK and Tajik students' responses prior to video testing as well as their responses afterwards was significant above the 95% level using the KS test (D-stat: 0.84, 0.79 and D-crit: 0.30 and

0.30, respectively).

### 3.2.3 Students' understanding of non-structural hazards

Figure 6 shows students' ability to identify non-structural hazards in example photographs. Non-structural earthquake hazards are caused by the furnishings and non-structural elements of a building (e.g., suspended ceilings and windows). In general, students from both groups identified non-structural hazards that are located above the ground (e.g., hanging television set and plant pots

stored above cabinets) and missed those near the floor (e.g., desks and chairs). Both groups demonstrated some knowledge of non-structural hazards found in typical school classrooms prior to video testing, and showed some improvement after video testing. However, only the difference between pre- and post-assessment responses by the UK students was statistically significant above the 95% level as indicated by the KS test (D-stat: 0.43, D-crit: 0.30).



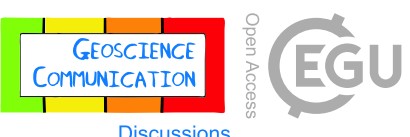

### 3.2.4 Earthquake prediction

Figure 7a shows students' responses to "Is it possible to know the exact timing of earthquakes before they occur?" While the majority of Tajik students indicate "no" in their responses before and after video watching (55% and 68%, respectively), UK students' responses are more evenly distributed between three categories of "no", "sometimes" and "I don't know" prior to video watching (36%, 33% and 28%, respectively) as well as after video testing (46% indicate "no", 38% indicate "sometimes"). After video watching, a notable percentage of Tajik students (21%) believes that it is possible to know the exact timing of an earthquake.

In their pre-assessment responses to whether it is possible to know where an earthquake can occur (Fig 7b), a significant percentage of both groups (47%: Tajik, 41%: UK) believe that is sometimes possible. After video watching, Tajik students' responses are divided almost evenly in four different categories: 26% indicate that it is possible to know where earthquakes can occur (category "yes") while 18% believe otherwise (category "no"), 21% indicate it is sometimes possible, and 26% give irrelevant answers. In contrast, the majority of UK students' post-assessment answers is "yes" to earthquake location prediction (62%) with "sometimes" indicated in 28% of responses. The differences between UK pre- and post-assessment responses is statistically significant above the 95% level as indicated by the McNemar test (chi-stat: 6.66, chi-crit: 3.84).

Additionally, we asked students an open-ended question: where in the world earthquakes occur most often? Based on their answers, we created five response categories (Fig 7c). A large percentage of students' responses from both groups include country names before and after watching the videos. UK students also included plate boundaries in their responses both before and after watching videos (21% and 40% respectively). In contrast, Tajik students made references to mountains/volcanoes (39% and 44%, respectively). These observations match students' responses to causes of earthquakes (Fig. 5). The differences between UK and Tajik post-assessment responses is statistically significant above the 95% level as indicated by the KS test (D-stat: 0.38, D-crit: 0.30).

## 4 Discussion

### 4.1 Factors influencing students' learning

The differences between students' pre- and post-assessment responses to survey questions were only significant above the 95% level when they were asked to indicate if it was possible to know where earthquakes happen (section 3.2.4), sketch a cross-section of the Earth (section 3.2.1), and identify non-structural hazards in example photographs (section 3.2.3) (Table 2). These results indicate that our paired teaching videos appear to change student views and understanding of some concepts including (i) Earth's interior for the Tajikistan group, (ii) earthquake location forecasting for both the Tajikistan and UK groups, and (iii) non-structural hazards for the UK group. However, our curriculum did not significantly change students' understanding of other concepts including causes of earthquakes and their relation to plate boundaries. Below, we discuss possible factors improving or hindering students' learning of these targeted topics.

*Interior of the Earth* - While a notable percentage of Tajik students' responses (50%) indicate an increase of at least 20% in their understanding of the Earth's interior, only four (out of 38) students demonstrated an advance understanding of the Earth's interior structure where scale is important (score 4-5). This is despite the fact that students were shown a diagram of a cross-section of the Earth (with all layers drawn to scale and labelled) by the video teacher, and participated in a classroom activity during which they compared the interior structure of a hard-boiled egg with that of the Earth. These concepts, however, were unfamiliar to Tajik



students prior to video testing and were not repeatedly reinforced during and after video implementation. It is important to note that more than 50% of Tajik students did not label the Earth's layers in both pre- and post-assessment surveys since this was not

specified by the question. However, the post-assessment data shows a 24% increase in the number of students who draw the interior of the Earth as concentric circles. Therefore, it is possible that students' understanding of the Earth's interior after video watching is underestimated. For the UK group, 46% (18 out of 39 students) showed no or naïve conceptual framework of the Earth's interior after video testing, with only 31% (12 students) showing some improvement. We therefore conclude that diagrams and simple analogies can bring some improvement to learning these concepts, but to make significant conceptual advances, concepts must be

revisited and reinforced repeatedly.

*Causes of Earthquakes* - The Earth's interior and Plate boundary videos did not increase students' understanding of earthquakes and associated processes. After video testing, only one Tajik student (out of 38) listed plate tectonics as the main cause of earthquakes. The relationship between plate tectonics and earthquakes was briefly mentioned by the video teacher in the Earth's

interior video. However, during the testing of the Plate boundary video, students used different datasets (including an earthquake map) to observe data behaviour near/at plate boundaries. It is possible that students' incomplete understanding of the Earth's interior structure (as described above) hindered their learning process associated with earthquakes as shown by Barrow and Haskins (1996). Another hindering factor could be the lack of previous exposure and concept reinforcement during/after video testing. Relating earthquakes to volcanoes and mountains is part of students' pre-existing knowledge as demonstrated in 39% to 50% of students'

pre-responses to Question 1 (i.e., What are the causes of earthquakes?) and Question 4 (i.e., Where in the world do earthquakes occur most often?), respectively. Students' pre-existing knowledge of why and where earthquakes occur was revisited and reinforced in the Plate boundary video and the accompanying classroom activities where students explored relations between distribution of volcanoes, topography, and earthquakes. Relating these processes to plate boundaries, however, was a new concept that was not reinforced. In contrast, nearly all UK students (35/39) connect earthquake occurrence with plate tectonics prior to

video watching. Because of their previous knowledge, it is difficult to assess the effectiveness of the plate boundary curricular activities conducted with this group.

*Earthquake prediction and forecasting* - The two closed survey questions that assessed students' views of earthquake prediction (Question 2 and 3) in terms of earthquake location and timing were not directly addressed by our curricular material. However,

these questions were selected to (1) assess students' current perception of and views on earthquake prediction and forecasting in general and (2) to find out whether learning about the Earth's interior and earthquake-related processes alone can change students' views of earthquake prediction and forecasting. This information is important since earthquake prediction/forecast, or lack thereof, may influence preparedness attitudes and behaviors in some communities. Prior to video watching, approximately 60% (23 Tajik students) said "no" to earthquake prediction (in terms of date and time) and "yes" or "sometimes" to earthquake forecasting (in

terms of location). In contrast, these values for the UK group are lower (36%, 14 students) and higher (77%, 30 students), respectively. The only significant difference (95% level) between students' pre- and post-answers was observed for the UK group for earthquake location forecasting. This may indicate that our curricular material (particularly plate boundary classroom activity), which emphasizes that earthquake locations are not random, was effective in changing students' understanding of earthquake location forecasting. A notable portion of Tajik students (26%), however, misinterpreted the survey question as evident in their

irrelevant responses to earthquake location forecasting (e.g., listing streets and schools as locations of earthquakes).

*Non-structural hazards* – For both student groups, our non-structural hazard video increased the students' ability to identify non-structural hazards in example classroom photos. However, the increase in hazard identification is significant (95% level) only for





the UK group. This could be due to lack of exposure to earthquake shaking during which non-structural elements of a building can

pose hazards to building occupants. Having experienced earthquakes and familiar with some hazards, Tajik students' give similar responses before and after video watching.

### 4.2 Teacher feedback

*UK teacher feedback-* Both teachers found the curricular material to be presented clearly, appropriately geared to the level of their

students, and valuable in helping students understand and learn the lesson content. However, there were differences in how they rated the materials in terms of relevance. In general, classroom activities carried out during video breaks were described as relevant by both teachers. However, both teachers found activities related to cost-benefit analysis of non-structural hazards in Video 3 to be irrelevant with low levels of student engagement when compared with other assignments in their class. One teacher explained, "[students] did not find the tasks that useful/applicable to life as we are not in an earthquake prone area." Similarly, another teacher

said, "[…] compared to the way we normally teach tectonic hazards, this (1 hr 40 minutes) felt like a lot of curriculum time on a relatively narrow aspect of the topic." There were some differences between teachers' opinions and experiences with video testing. For example, while one teacher found the egg analogy to be a "very relevant" activity and described students' level of engagement with this activity as "a great deal more" than other classroom assignments, the other teacher recommended omitting this activity, "[…] I think they could have completed this task more effectively without being given and cutting up the egg (which took up

time and were a distraction)." Teachers played the entire videos in the classroom with the exception of the plate boundary video for which one teacher played only two out of five video segments, and skipped related classroom activities. The same teacher described students' level of engagement to below average for the two activities that were carried out.

*Tajik teacher feedback* - Video testing in Tajikistan had to be conducted by the lead author (a geosciences researcher and teacher)

due to the local teachers' lack of interest and/or confidence in having the required teaching skills and knowledge. This was despite making teaching materials accessible and available in the local language in advance of video testing, and holding an informal meeting with local teachers during which they were introduced to paired teaching and lesson content. To further support and encourage the local teachers to participate in video testing, a trained teaching assistant familiar with the paired-teaching videos was made available. The school principal was supportive of teachers' participation but did not require it. Due to limited time and

resources, we had to carry out the video testing with little contribution from local teachers. The teacher feedback discussed here, therefore, reflects our experience with video testing.

With the exception of the plate boundary video, the level of content covered in both the Earth's interior and non-structural hazards videos were appropriate for Tajik students. During the plate boundary video testing, most students were not only challenged by the

lesson content (e.g., understanding maps, data relationships and classification), but also struggled with understanding and following the pedagogical approach (e.g., collaborative learning) required for classroom activities. This lowered the class pace, and therefore, the video was played partially, with some classroom activities omitted. Students' level of engagement was the lowest for the plate boundary video, and the highest for the non-structural hazard video which focused on a topic most student could relate to (identifying hazards in their own classroom). The level of student excitement with lessons, however, was the highest when watching

the Earth's interior video which was taught by a UK-based video-teacher. The excitement was less for the subsequent videos which were taught by the lead author who had to act as both the video and the in-class teachers. The video testing was also affected by the choice of space and place for curriculum implementation. For example, the Earth's interior video was tested during normal school hours but outside the classroom environment. For this lesson, the stage in the school theatre was selected by school



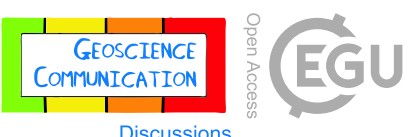

authorities because the equipment required (e.g., projector, screen, and sound system) was only available for use there. For the remaining two videos, the teaching space was changed to a regular classroom and a computer lab. While the latter restricted students' movements and group formation for activities, the former provided a familiar and flexible space where students and in-class teacher could easily rearrange chairs and tables according to their needs.

Teacher feedback highlights some important differences and similarities during classroom video implementation in two geographically and culturally different parts of the world. As the above observations show, students' experience with earthquakes and related hazards influence their level of engagement with lesson content. While technological shortcomings and ineffective classroom space and management significantly challenged video testing in Tajikistan, these factors were non-existent in the UK school. The poor classroom management in the Tajik school was exacerbated by the lack of local teachers' involvement which led them to bring in a new teacher (lead author) who was unfamiliar with the specifics of the classroom culture. Despite these differences, both the UK and Tajik teachers acted flexibly with video testing by skipping video segments irrelevant to students' lives or shifting to printed lesson plans and other resources when technology failed. Taken together, to maximize the impact of our paired teaching approach to earthquake education, we suggest to exercise flexibility when using our videos and contextualize video content and learning activities to increase their relevance. To encourage and assist teachers with lesson preparation, we plan to improve our "teacher segment" by creating a guide for each video in a printable format with descriptions of activities covered in each segment as requested by UK teachers.

### 4.3 Lessons Learned

This study brought together a group of early-career Earth scientists to develop and test a series of DRR educational materials to support school teachers with earthquake education worldwide. The paired teaching technique provided an effective means for these scientists to connect and co-teach curricular materials with school teachers that might have not been able to invite, host, and co-teach lessons with them. In the case of Tajikistan, these lessons were implemented in a country that lacks the economic resources to develop and promote a school-based geohazards education (Mohadjer et al., 2010) despite a clear need for it. Therefore, our paired teaching video lessons can be an effective way for Earth scientists worldwide to engage with school teachers regardless of their locations (assuming language and technological barriers are addressed as it was the case in this study). Below, we discuss some of the lessons learned in this study, to increase the impact of future Earth sciences education and outreach efforts taken by Earth scientists worldwide.

*Classroom culture-* An effective DRR-related school-based education is one that is contextualized to meet local needs, and carried out in the cultural context that surrounds the implementation of the curriculum. Students' and teacher's needs and goals, local constrains and schools' pedagogical values are some of the factors that shape a classroom culture. While some topics such as Earth's interior and plate tectonics are often covered to varying degrees in most schools around the world, knowing how to identify and fix non-structural hazards related to earthquakes might not be as relevant to schools located in non-seismic regions. Similarly, content taught using pedagogical approaches that are unfamiliar to some teachers and students can hinder effective learning. For example, most of the learning activities in this study were based on cooperative learning (e.g., jigsaw concept) and involved group discussions where everyone's input was encouraged. Unfamiliar with this approach, these activities appeared to be unstructured to most Tajik students, leaving some uncomfortable with sharing their opinions. When developing curricular material, teachers' and students' involvement are key to ensuring an appropriate selection of content and pedagogical approaches. This can be achieved through informal classroom observation and discussions of goals and pedagogical expectations with classroom teachers and students.

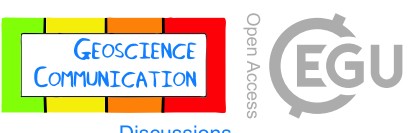

*Curriculum evaluation-* Nearly all Earth scientists agree that public outreach is important. However, many of those who practice science education and outreach do not always evaluate their work rigorously, and even fewer publish and share their results in peer-reviewed journals. By testing our videos in school classrooms located in different countries (UK and Tajikistan), we were able to assess the effectiveness of our educational materials and identify potential factors that influence learning. However, the assessment of our evaluation strategy revealed several issues. The completion of the pre-assessment survey was an unfamiliar

assignment to most Tajik students. Many students perceived it as an exam and strived for correct answers as opposed to freely sharing their existing knowledge. Despite gaining new knowledge, some Tajik students gave identical responses to pre- and post-assessment questions in order to be consistent as opposed to be correct. In addition, after video testing, we asked both the UK and Tajik students to make models of the Earth's interior using Playdough that came in different colors. However, most Tajik students blended the colors to form new ones as opposed to using different colors to indicate different layers of the Earth. The Playdough

for UK students went missing before students had a chance to use it. Taken together, an effective evaluation strategy should take into account students' familiarity with data collection methods to ensure students understand what they are being asked to do and why.

## 5 Conclusion

In this study, we created ten geoscience video lessons focusing on earthquake science and hazards. The paired teaching approach

was used to engage geoscientists as video teachers who introduce and discuss concepts in brief video segments. In between segments, these concepts were explored through hands-on activities under the guidance of an in-class teacher. We tested three videos (Earth's interior, Plate boundaries, and Non-structural hazards) with a total of 77 students (12-14 years of age) from schools in Dushanbe (Tajikistan) and London (United Kingdom). Our analysis of students' pre- and post-assessment responses to survey questions indicate: (1) students' pre-existing conception about the causes of earthquakes is difficult to modify if new concepts are

not repeatedly reinforced, and (2) students' incomplete understanding of the Earth's interior hinders their learning process associated with earthquakes. Comparison of results from the UK and Tajikistan groups reveal significant differences between students' views on the Earth's interior and why and where earthquakes generally occur. Possible factors influencing students' learning are those related to students' own experience with earthquakes, pre-existing knowledge, and unfamiliarity with some content (e.g., data maps) and pedagogical approaches (e.g., collaborative learning). These factors should be taken into account in

order to maximize students' learning during paired teaching.

Despite documenting an increase in students' understanding of some concepts covered in the tested videos, the effectiveness of our entire video series cannot be fully assessed without furthering testing. This is because the series follows a stepwise approach to increasing students' understanding of earthquake science and hazards, with later lessons in the series building on topics covered in

earlier lessons. This approach allows for reinforcement of some difficult concepts. However, excessive workload, restrictive curriculum, and increased pressure to achieve good result limit teachers' decisions to use the entire series which includes ten videos (total of 10-20 hours of classroom time). Therefore, we recommend selecting and using video segments that are relevant to (and can enhance) the teaching of specific topics covered by an already existing curriculum. In addition, our videos can serve as a resource for teachers who cannot easily arrange for an in-person or a virtual live session between their students and an Earth

scientist.



Geoscience and natural hazard researchers' contribution to developing resilient communities is often through engagement in disaster risk reduction (Gill et al., in review). We hope that lessons learned in this study can benefit the scientific and wider DRR community by highlighting some of the key factors that influence the teaching and learning of geo-hazard content. There is already

a wide range of tools and resources developed by the geo-hazard community to ensure meaningful access to scientific information relevant to DRR. Examples include Hazard Ready (a hazard preparedness web application developed for specific cities in the United States), the Central Asia Geohazard database (a searchable repository of active fault and earthquake information by Mohadjer et al., 2016) and various earthquake data products created by the Global Earthquake Model (GEM) team (e.g., the Global Seismic Hazard Map shown in Fig. 1). These resources, if contextualized appropriately, can be effectively incorporated into DRR

educational materials (e.g., our paired teaching videos, animations, and exercises) used with the K-12 and higher education communities.

**Data availability**

All video files are archived and available for download at Technische Informations Bibliothek (TIB) Av-Portal (https://av.tib.eu/media/47600). In addition, videos are available for view at the YouTube channel of the European Geosciences

Union: https://www.youtube.com/user/EuroGeosciencesUnion.

**Author contribution**

SM and SGM were responsible for video production, direction, editing and planning the study. Videos evaluated in this study were presented by MK and SM. Classroom testing of videos in Tajikistan was coordinated by AI and led by SM. MK and SG arranged and coordinated the video testing in the UK school. Data analysis was done by SM and SGM. All authors contributed to manuscript

preparation.

**Completing interests**

The authors declare that they have no conflict of interest.

**Acknowledgement**

We thank Ruth Amey, Jessica Starke, Lewis Mitchell, Reinhard Drews, and Matthias Nettesheim for helping with video

presentation and filming. We are grateful to Joel Gill, Bruce Malamud, Jordan-Cyrus Seyedi, Charlotte Jackson, and Faith Taylor for coordinating and assisting with video filming in London. We also thank Richard Larson and Elizabeth Murray for helpful discussions. We would like to express our appreciation to the principals, teachers and staff members of schools selected for this study, and for ongoing collaboration with the Institute of Geology, Earthquake Engineering and Seismology in Tajikistan. This work was supported by a European Geosciences Union public engagement grant to S.M. and by the CAME II project bundle

CaTeNA of the German Federal Ministry of Education and Research (BMBF support code 03G0878D to T.A.E.).



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




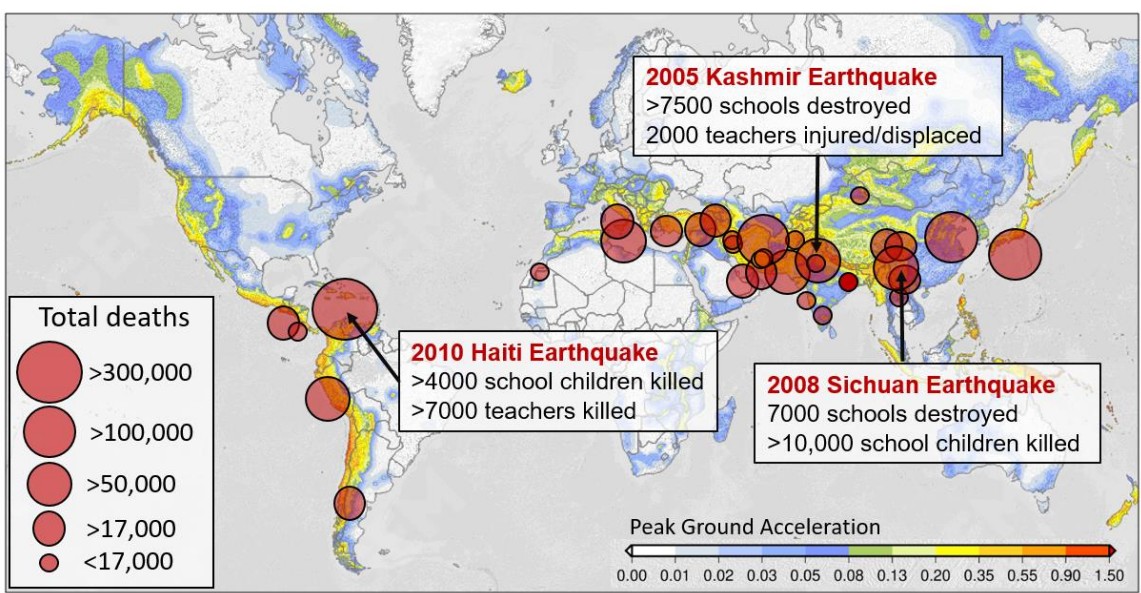

**Figure 1. Significant earthquakes from NOAA database superimposed on the Global Seismic Hazard map of Pegani et al. (2018). Impact details of recent earthquakes discussed in text are shown in white boxes.**

















Figure 2. Paired teaching (teaching-duet) pedagogy






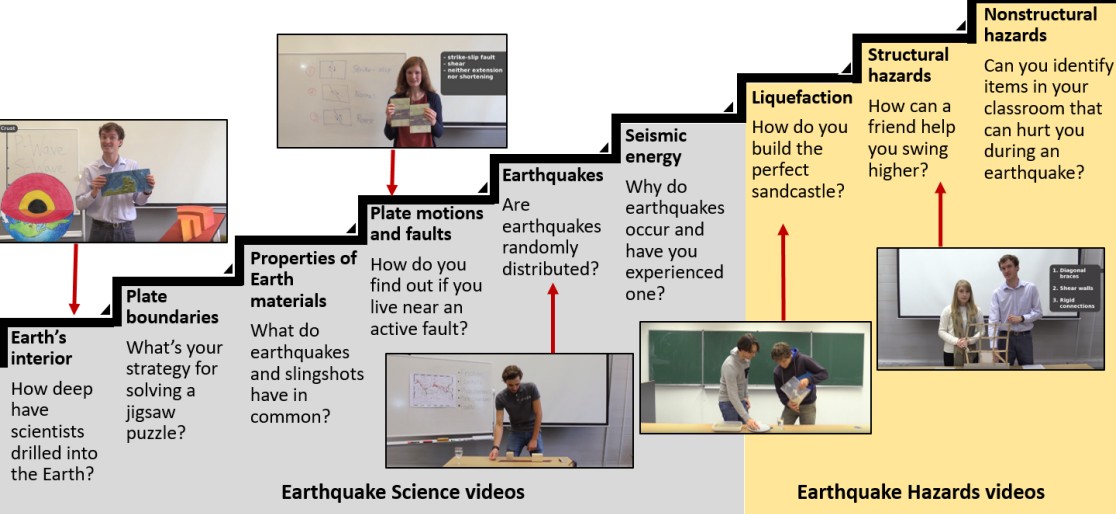

Figure 3. Stepwise earthquake education curriculum (modified from Mohadjer et al., 2010)







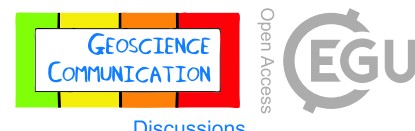


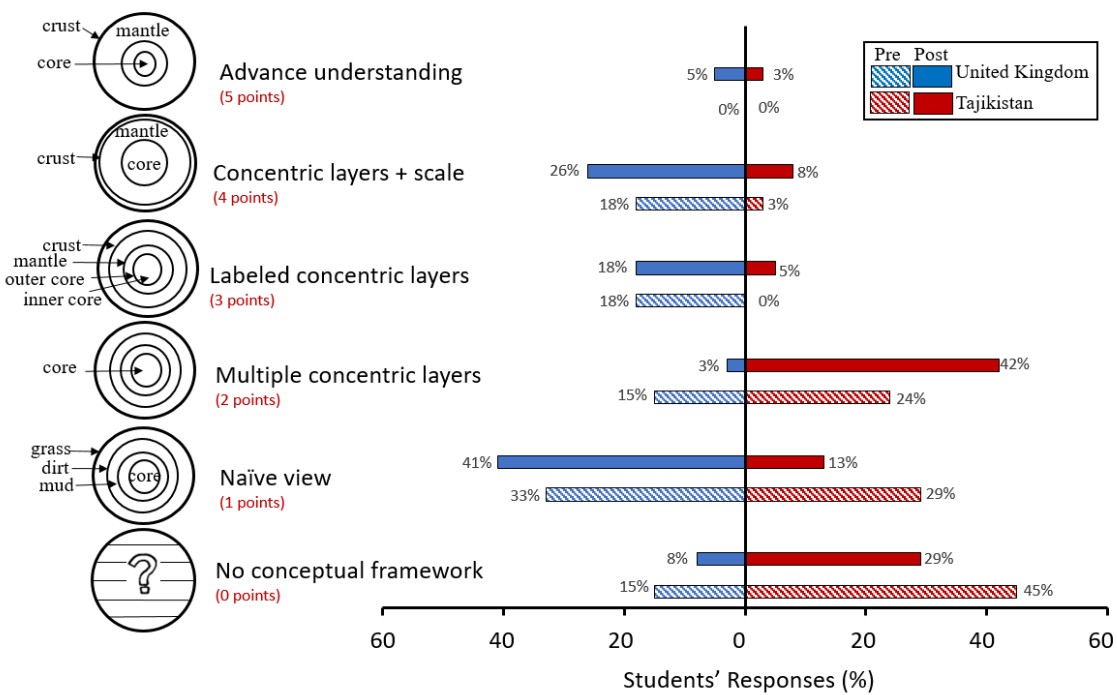

**Figure 4: Students' understanding of Earth's interior (right) when asked to sketch a cross section of the Earth, evaluation rubric after Steer et al., 2005 (left)**










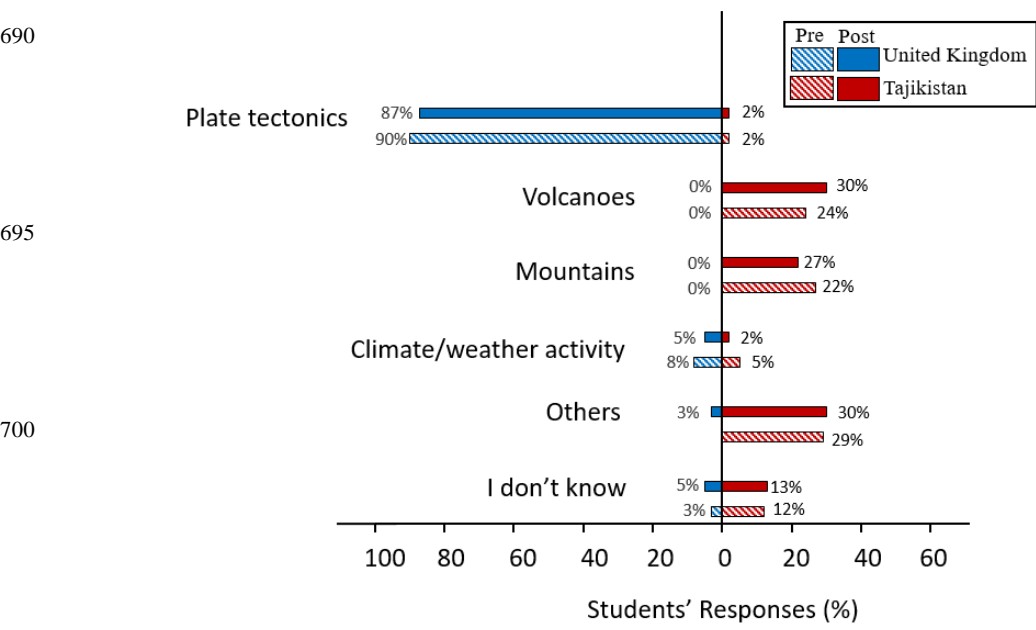




**Figure 5: Causes of earthquake according to students.**







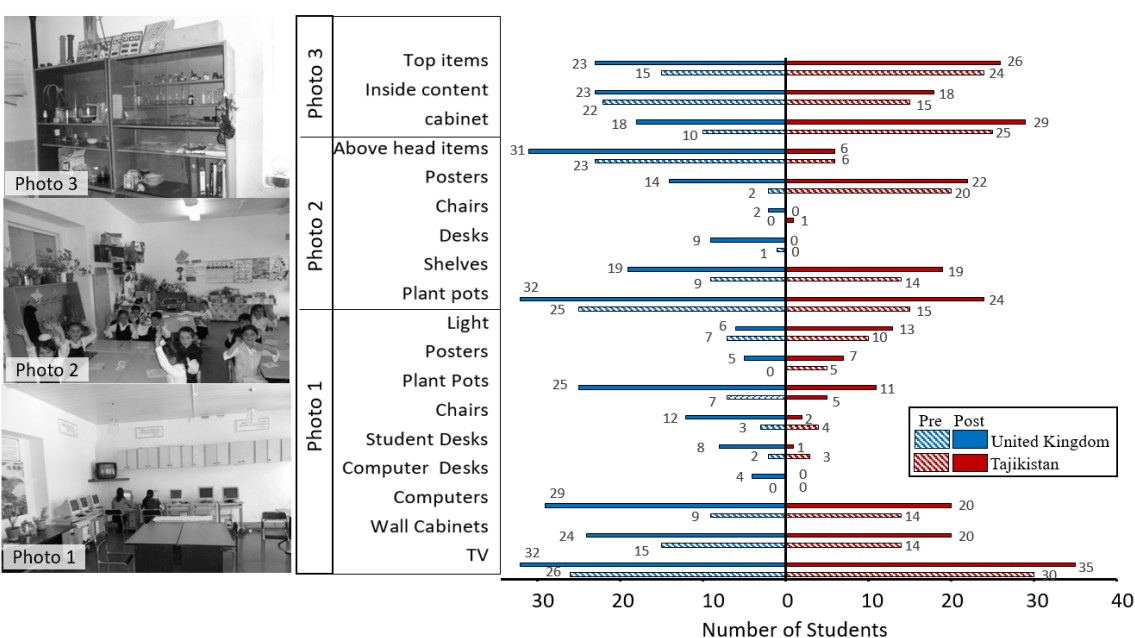

**Figure 6: Students' understanding of non-structural hazards. (from left to right) Photographs used in questionnaires for non-structural hazard identification, items identified correctly as non-structural hazards, and pre-/post-assessment results. Photo credit: AKDN/DRMI, 2011 (www.akdn.org).**






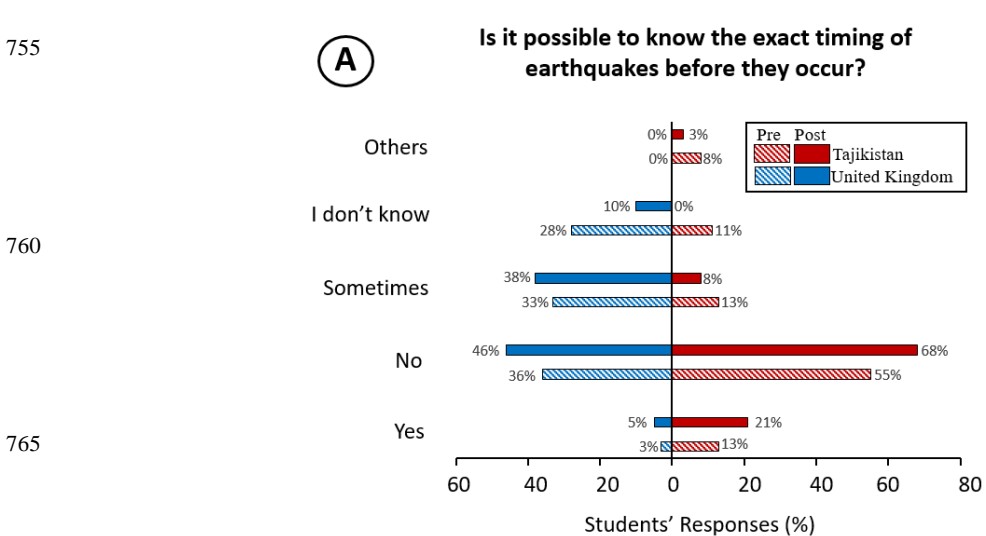




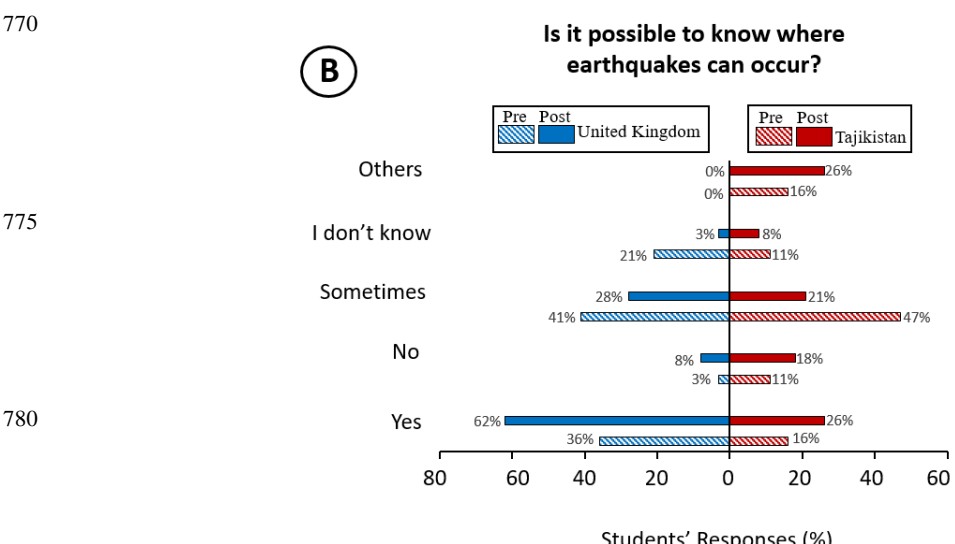








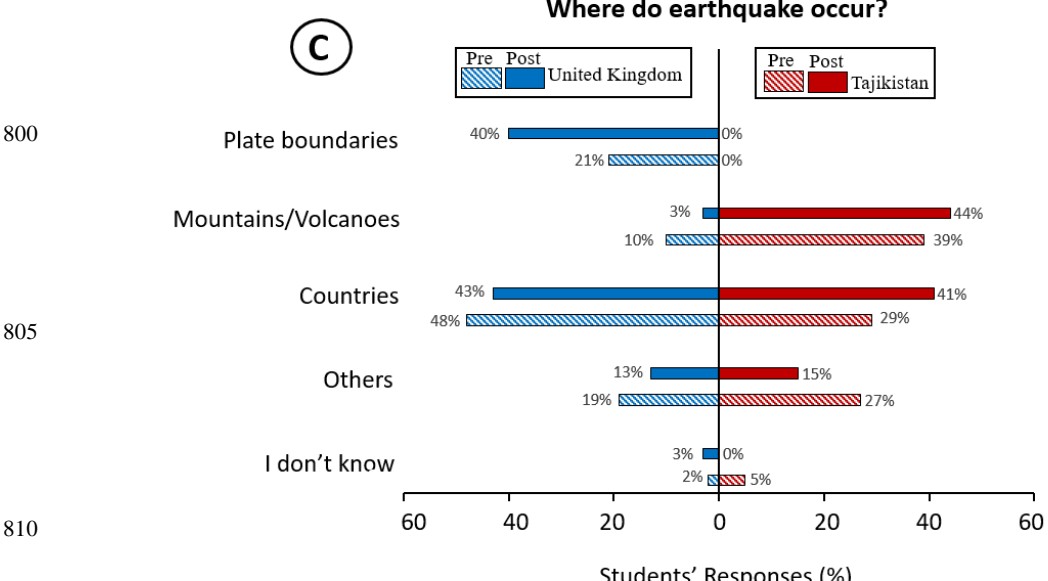

Figure 7: Students' understanding of earthquake prediction including earthquake timing (a), location (b) and where in the world earthquake occur most often (c).












| Title | Teaching approach | Topics covered | Main classroom activities | DOI & video length |
|---|---|---|---|---|
| **Journey to the center of the Earth**: Earth's interior and plate tectonics | Conceptual model, interactive lecture demonstrations, use of analogies and other visualize aids | Earth's internal layers, methods for investigating Earth's interior including seismic waves | Using an hard-boiled egg as a scale model for the layers of the Earth; using Slinky toys to model seismic waves; discussing model limitations | https://doi.org/10.5446/47600 16 min |
| **Living on the edge**: Discovering plate boundaries | Cooperative learning (jigsaws), role playing, and data-driven exercises | Linking plate boundary processes to scientific observations, and the scientific method | Using data maps (e.g., earthquakes, volcanoes, seafloor age, topography) to investigate plate tectonic boundary processes | https://doi.org/10.5446/47601 24 min |
| **Soft rocks and hard liquids**: Properties of Earth materials | Interactive lecture demonstrations (using everyday objects) | Why and how materials deform; what controls deformation and energy transfer | Applying force to everyday objects, observing and identifying factors influencing their behavior | https://doi.org/10.5446/47700 13 min |
| **Do you know your faults?** Plate motions and faults | Use of models, teaching with visualizations (photos) and art | Causes and types of plate tectonic stress and resulting strain, mechanics of fault rupture | Using a dough to model Earth's crust under stress, building different fault models using pieces of cardboard | https://doi.org/10.5446/47701 14 min |
| **What causes that Rock'n'Roll?** The Earthquake machine | Classroom experiments (building and operating models) | Earthquake mechanisms, stick-slip motion, and earthquake prediction | Operating a mechanical model of a fault to observe fault motion during an earthquake, exploring the effects of several variables, and discussing model limitations | https://doi.org/10.5446/47702 12 min |
| **Rocking, rolling and bouncing**: How do earthquakes move the Earth? | Interactive hands-on demonstrations, use of models and animations | Waves as energy transfer, seismic waves and how they travel through different materials | Using a setup to show how seismic waves can travel through different materials, modeling seismic waves using Slinky toys and human bodies (human waves) | https://doi.org/10.5446/47703 21 min |
| **Flow with the sand**: Introduction to soil liquefaction | Classroom experiments, use of models and visualizations | Soil saturation and consolidation, causes of soil liquefaction and mitigation measures | Building a liquefaction model and using a shake table to test its response to shaking | https://doi.org/10.5446/47704 13 min |
| **Safe or unsafe**: Nonstructural hazards during earthquakes | Place-based learning and role playing | Nonstructural hazards identification and mitigation, rapid visual screening method, repair cost analysis | Identifying nonstructural hazards in school classrooms and discussing and proposing mitigation strategies | https://doi.org/10.5446/47705 20 min |
| **On shaky ground**: Structural hazards during earthquakes (Part I) | Classroom experiments | Introduction to shake table; how different materials respond to different loads | Constructing building models and testing them on a shake table, discussing model limitations | https://doi.org/10.5446/47706 13 min |
| **On shaky ground**: Structural hazards during earthquakes (Part II) | Classroom experiements, group discussions | Earthquake engineering of buildings, frequency, natural frequency and resonance | Constructing and testing building models on a shake table, modifying them to reduce resonance, and discussing model limitations | https://doi.org/10.5446/47707 16 min |


**Table 1.** Summary of the teaching approaches, topics covered, and classroom activities for each video including video duration and its digital object identifier (doi).
