# Peer review of "Using paired teaching for earthquake education in schools"

_Geoscience Communication, 2020_

## Referee Comment (RC1) · Anonymous Referee #1 · 20 Nov 2020

Referee's Comments on the Paper: "Using Paired Teaching for Earthquake Education in Schools," Mohadjer, et al. November 20, 2020

This is a well-written paper describing in-classroom use of the "teaching-duet" or "paired-teaching" methods put forward by MIT BLOSSOMS https://blossoms.mit.edu. The paper reports on the creation of ten instructional geoscience videos at the level of middle school and/or high school, to teach the students (in paired-teaching mode) the science of earthquakes and the engineering design of earthquake preparedness and response to reduce loss of life and property. These topics are especially relevant to students in Tajikistan, one of the worlds' most earthquake-prone countries.

The referee believes that the paper is deserving of publication, after considering some modest revisions.

1. The paper, in the middle, "goes into the weeds" on lots of little in-class details when the reader hungers for the "bottom line," i.e., Did it work? Yes or No? And why? Suggest that lots of these minutia details be relegated to an appendix, perhaps shortening the main text by 50% or so. The main text can summarize "the weeds." 2. "...the local language (Tajik) was used in teaching and in all written and media materials." Does this mean that the paired teaching videos were available in Tajik as well as in English? The manuscript is unclear on this. It states that the live discussions the of students and teachers were in Tajik, but is unclear on the videos. If the students in Dushanbe were shown English-spoken videos, then the lack of before-and-after knowledge improvement is most understandable. 3. The key message is the "underwhelming" amount of new learning, especially in Dushanbe. This reviewer was unable to discern how much of this is due to language (were the videos in English?) and how much to culture. Interactive classrooms are quite novel in most countries – where the tradition is that the teacher is undisputed leader and the students dutifully follow. Perhaps it would have been better to start the students on paired-learning materials on more traditional STEM subjects like math or basic science. But this is easier said than done due to lack of video materials in the local language. 4. The failure of teachers in Dushanbe to serve as in-classroom teachers in the paired-teaching mode may be cultural and/or due to lack of training in this new pedagogical model. Please discuss. 5. The recently renovated school in Dushanbe: Was it designed with the latest protections for earthquakes, with their destructive threats to life and limb? Answering that question would have been a nice addition to class discussions, especially if the answer was very positive – that is, substantial structural improvements and student training improvements.

---

## Referee Comment (RC2) · Anna Hicks (Referee) · 11 Dec 2020

This is a paper clearly written by geoscience enthusiasts with a passion for science communication, geoscience education and outreach. I applaud the authors' efforts to develop the videos, and their endeavours to evaluate the effectiveness (on knowledge gain/change) of the information in the videos with students in two culturally diverse countries. Evaluation of science communication is important and so often overlooked. For this alone, the paper is a valuable contribution to the literature.

The videos are basic, and have been seemingly produced with limited (or no) budget, which is deserving of praise. However, in the future I would welcome developed versions which build on an evidence base to address aspects such as cultural rele-

vance (e.g. words, images, language), local context, curriculum timing, trusted and identifiable presenters etc. There are numerous avenues this research could take, and I encourage the authors to explore opportunities to develop this area of study. For example, as Reviewer 1 has also raised, the authors have missed an opportunity to reflect on and share experiences of the cultural aspects of this study.

For my review of this work, I have provided some moderate-to major and minor points that I recommend the authors address. I have also provided some points for reflection, which may improve the manuscript (and future research) if the authors wish to address these. If these moderate revisions are made, I recommend this article for publication.

Major points to address:

1. Teaching experience: The authors talk about incorporating teaching strategies into the videos but provide no information about the level of teaching knowledge and experience of those developing and presenting the videos. Are any of the authors teachers or researchers in education? They are all based in geoscience departments, but it is not clear about their teaching experience. I believe the lead author may have considerable experience. That being said, were teachers consulted before making the videos? Did the authors understand the curriculum in both countries and seek to co-develop these videos with the teachers and/or curriculum developers with whom they engaged? How did the authors know that earthquake education in schools needed supplementing? These are important reflections that should be in the paper. Beyond seeing making videos as a 'good idea' to them, can the authors demonstrate evidence for need?

The study follows MIT Blossoms protocols and 5E Instructional Model but the authors do not reflect on this in the lessons learned. It therefore comes across as a retrospective 'shoehorning' into a published methodological approach. I was also confused about whether 5E lessons plans or plans developed by Mohadjer (2010) were used. Please clarify in section 2.1.

2. Ethics and consent: Where are the ethics assessments and the consent forms for

this study? Working with minors and collecting data from them requires consent and ethical assessments. How will the data be used and stored? Could the authors urgently provide some information about this in accordance with GDPR.

3. Cultural aspects: Please could the authors provide some reflections on the cultural dimensions of this study? Empirical evidence is fine. For me, observations would be as useful, if not more so, than the numbers! I am intrigued for example about the student responses to the causes of earthquakes between the countries. Why haven't the authors explored this? More reflections on why the Tajik teachers did not engage would also be useful for readers.

4. Statistical information: The statistics are distracting and not really that helpful for the reader. Whilst I appreciate the authors are trying to demonstrate a rigorous evaluation of the videos, the reader is yearning for an interpretation and discussion. Much of the statistical information could be moved to an appendix, or tabulated simply.

Major points for reflection - An opportunity has been missed to develop science communication theory or incorporate social scientific methods into the study. - The authors throw around the word significant in the paper, but care should be taken as to whether it really is or not. - I like that the videos were broken up into segments, but it would have been beneficial to receive feedback from the teachers and students as to the desired length of the videos – they seem pretty long as a whole (12-24 minutes per video – 10-20 hours teaching time in total) - The study seems to be more focused on evaluating knowledge gain or change and has failed to uncover what it actually was that caused that knowledge change – was it the words, the presentation, the models, the presenter? If the authors have this information, it would be useful to include, or even provide some observations and reflections that might lead to further study. - Regarding the students feeling that the questionnaire was an exam, this is absolutely my experience too, particularly in areas where cultural differences between the students and the visitor are quite obvious. It would be really useful for the authors to reflect a bit on this, and ideally put it in the paper – might it be related to perceived or actual power,

positionality, cultural norms etc?

Minor points to address: Line 10 The opening line of the abstract is conjecture. Suggest a rewrite. Line 13 "coupled with activities carried out by local classroom teachers". Suggest a rewrite – the teachers aren't carrying out the activities, and what do you mean by local classroom? Isn't teachers enough? Line 21 I'd suggest replacing views with knowledge – you weren't evaluating their opinions. Although that would have been interesting. Line 25 What makes a classroom culture suitable? Who says so? Suggest a rewrite. Line 46 empowerment of communities Line 66 Be consistent with language – choose either teaching-duet or paired teaching Line 75 Typo on through Line 87 How did you develop an iterative process? There did not seem to be any iteration in the UK example – did the video teacher actually talk to the teacher in-situ and tweak approaches? In the Tajik example the video teacher was the in-situ teacher, so again, no iteration. Line 128 How do you know that the teacher continues to keep the students interested? Line 160 You collected data on the first and last days of video implementation but the testing periods were very different between the UK and the Tajik studies (50 and 5 days respectively). Can you reflect a bit on this, maybe in a later section? Line 214 Section 2, not 3. Line 293 How can you have a 20% increase in student understanding of the Earth's interior. Please clarify or omit this. Line 303 Might be a personal thing, but the word naïve feels a belittling – can the authors just say a basic conceptual framework? Lines 328-29 I like where the authors have put numbers of students in brackets, in really helps the reader. Consistency with this throughout the paper would be welcome. Lines 397 I'd argue that the authors have not produced a series of DRR educational materials, only a few related to DRR and most were about hazard education Line 409 constraints (typo) Line 411 Fix non-structural hazards? Could a more appropriate word be found? Line 411 non-seismically active regions Line 420 "Nearly all Earth scientists agree that public outreach is important." How do the authors know this? Where's the evidence/reference? Otherwise just conjecture. Line 460 I'd suggest changing the word ensure to support. Line 460-466 These examples are ok, but are they really related to K-12 and higher education? How about researching some

serious games such as Earth Girl and Hazagora?

---

## Author Comment (AC1) · 5 Feb 2021

We thank Anna Hicks for taking the time to review the manuscript (**gc-2020-43**: *Using paired teaching for earthquake education in schools*). We have updated the manuscript as suggested. The detailed response is shown below in **blue**.

1. Teaching experience: The authors talk about incorporating teaching strategies into the videos but provide no information about the level of teaching knowledge and experience of those developing and presenting the videos. Are any of the authors teachers or researchers in education? They are all based in geoscience departments, but it is not clear about their teaching experience. I believe the lead author may have considerable experience. That being said, were teachers consulted before making the videos? Did the authors understand the curriculum in both countries and seek to co-develop these videos with the teachers and/or curriculum developers with whom they engaged? How did the authors know that earthquake education in schools needed supplementing? These are important reflections that should be in the paper. Beyond seeing making videos as a 'good idea' to them, can the authors demonstrate evidence for need? The study follows MIT Blossoms protocols and 5E Instructional Model but the authors do not reflect on this in the lessons learned. It therefore comes across as a retrospective 'shoehorning' into a published methodological approach. I was also confused about whether 5E lessons plans or plans developed by Mohadjer (2010) were used. Please clarify in section 2.1.

**Response:** Thank you for these excellent points. We have revised the manuscript text to address these concerns. Below, we explain each concern separately.

Authors' experiences - All video lessons were developed under the guidance of the lead author (with over two decades of experience related to K-12 science education and outreach), with input from the MIT BLOSSOMS team including faculty and teachers who have published >100 STEM videos following the paired teaching technique. Other authors (Matthew Kemp, Sophie Gill, and Sebastian Mutz) are also experienced with developing and teaching STEM related educational activities in both formal and informal settings including K-12 schools, museums, and public outreach activities conducted by universities in the UK and Germany. Finally, the video scripts and all the classroom activities that are done as part of these videos, were adapted from published lesson plans (Mohadjer et al., 2010). These lesson plans have been tested, evaluated and improved by hundreds of teacher volunteers (members of Teachers Without Borders) across China, Tajikistan, Afghanistan, India, and Haiti. The video modules described in the manuscript are modified from these existing (and already evaluated) lesson plans, but differ in format and pedagogical approach (i.e., they're published as video lessons using the scientist/teacher paired teaching technique).

On supplementing the curriculum – Our video lessons are not intended to replace an existing curriculum but rather to support the teaching of concepts related to earthquake science, hazards, and safety through interactive, hands-on activities and discussions that teach students to think critically. These discussions/activities are ignited and guided by a guest scientist and the in-class teacher. It is important to note that our video lessons are developed for the global school community, and do not target or adhere to the educational standards of a specific region or country. However, we encourage (and provide examples) to interested teachers on how to contextualize the content according to their local environment. We do this in the video segment titled "Teacher's Guide" at the end of each video. To clarify this, we have modified the manuscript (section 2.1.1).

In addition, we have observed that in both countries, topics related to natural hazards are textbook-driven and often discussed briefly, with no/little active learning exercises to encourage critical thinking. Particularly in Tajikistan, rote learning and classroom management predominate. Furthermore, inviting local/international scientists to co-teach with classroom teachers is rare in Tajikistan and some parts of the UK. Our video lessons,

therefore, are not only an opportunity for igniting active learning in schools, but also allow for "virtually" bringing scientists into school classrooms without additional costs/resources. Concerning the 5E lesson plans (Mohadjer et al., 2010), we have modified the manuscript to clarify the confusion raised by the reviewer. The content of the 5E lesson plans were used only as a guide by the video teachers when developing the video lessons. Therefore, the 5E method was not used in the production of the video lessons (hence no discussion of it in the manuscript), but the content of the lesson plans was adapted, modified, and improved by the video teachers (under the guidance of the lead author) to make them suitable for paired teaching. Therefore, the main pedagogical approach used in this manuscript is the paired teaching method which is discussed extensively in the method section (section 2.1) as well as in Larson and Murray (2017) which is referenced in the manuscript. The reason the 5E instructional model was mentioned in this manuscript (section 2.1) is to differentiate the video lessons (focus of this manuscript) from the lesson plans of Mohadjer et al.

2. Ethics and consent: Where are the ethics assessments and the consent forms for this study? Working with minors and collecting data from them requires consent and ethical assessments. How will the data be used and stored? Could the authors urgently provide some information about this in accordance with GDPR.

**Response**: Ethical approval for this study was sought and received from the participating schools and institutional partners that coordinated this effort in 2018-2019. The ethical procedures were designed to adhere to current standards of assent and consent regarding in-school research and to provide participants with anonymity. As a low-risk, school-based study focused on learning outcomes from regular teaching activities, the consent of the school-based stakeholders, i.e. the principals and teachers, were considered sufficient to proceed with the pre- and post-tests. All participation was voluntary, and students were given the opportunity to assent or refuse participation at both the pre- and post-test points in time. The pre- and post-tests were anonymous with students creating their own codenames, which were known only to them, that were used to match the pre- and post-tests for analysis purposes. No sensitive or identifying information was collected, and the anonymous data are stored in a secure location within the European Union that is password protected, in accordance with GDPR regulations. The data will be destroyed upon completion of this research project.

3. Cultural aspects: Please could the authors provide some reflections on the cultural dimensions of this study? Empirical evidence is fine. For me, observations would be as useful, if not more so, than the numbers! I am intrigued for example about the student responses to the causes of earthquakes between the countries. Why haven't the authors explored this? More reflections on why the Tajik teachers did not engage would also be useful for readers.

**Response**: Thank you for these important points. We discuss each separately below. The manuscript text is also modified to include this reflection:

Cultural dimensions - This study was not designed to assess the cultural differences in students' understanding of earthquakes. However, our earlier work (Mohadjer et al., 2010) gave additional insights into the cultural dimensions of earthquake knowledge among school children in Dushanbe. Unlike our 2010 study, we documented no mention of myths, legends, or religious explanations in Tajik students' responses to our interview questions probing them on earthquake causes and preparedness measures. This could be due to (1) different data collection methods (anonymous survey questionnaires vs individual/group interviews), (2) differences in students age/grade level (the 2010 participants were 2-3 years older than the 2018 participants), and/or (3) may reflect changing earthquake perceptions over the last decade. As for differences between students' responses from UK and Tajikistan, it is clear that the Tajik students (unlike those from UK) lacked the correct scientific terminologies

(e.g., plate tectonics), but could associated processes such as mountain building and volcanism to causes/locations of earthquakes. Living in an earthquake-prone country that is mountainous could be one factor explaining their answers. This was discussed in section 4 under "causes of earthquakes".

Low teacher engagement - Thank you for this point which was also raised by another reviewer. As for why the Tajik teachers did not engage, we believe this is due to their unfamiliarity with an interactive/collaborative classroom culture, and their level of comfort operating within it. Below, we explain what we learned from this observation. The manuscript text is also modified to include this reflection:

The paired teaching video lessons are designed to be a complete resource. They include video segments, teacher's guides, downloadable handouts and lists of other resources relevant to the topic. Therefore, no teacher training should be needed in order for teachers to use these videos. However, teachers are encouraged to view the videos and familiarize themselves with the content before using them in their classrooms. This study, however, reveals that these videos may not be seen as a complete resource by some teachers. While the UK teachers tested the videos with minimal input from video creators, teachers in Tajikistan asked to observe classroom testing of the videos. This request was made despite the fact that the teachers were offered (i) training to deliver the videos, and/or (ii) the option to co-teach the video lesson with experienced instructors. The textbook-based classroom culture (typical of schools in Tajikistan) may explain why Tajik teachers did not want to actively engage in video testing. The study, therefore, shows that the paired teaching pedagogy is not a "one size fits all" teaching approach, and depends on the classroom culture and teacher's comfort operating within it.

4. Statistical information: The statistics are distracting and not really that helpful for the reader. Whilst I appreciate the authors are trying to demonstrate a rigorous evaluation of the videos, the reader is yearning for an interpretation and discussion. Much of the statistical information could be moved to an appendix, or tabulated simply.

**Response:** Thank you for the comment. The statistical test results are included in a supplementary table (Table 2b). However, since we used statistical tests to compare results across and in between each group (UK vs Tajikistan) and to determine if differences observed were significant, we think including this information will help some readers. To increase manuscript readability, statistical information appears consistently at the end of each subsection (e.g., 3.2.1-3.2.4) in the result section, and are not mixed with the description/discussion of results. We hope this is a satisfactory answer.

Major points for reflection - An opportunity has been missed to develop science communication theory or incorporate social scientific methods into the study. - The authors throw around the word significant in the paper, but care should be taken as to whether it really is or not. - I like that the videos were broken up into segments, but it would have been beneficial to receive feedback from the teachers and students as to the desired length of the videos – they seem pretty long as a whole (12-24 minutes per video – 10-20 hours teaching time in total) - The study seems to be more focused on evaluating knowledge gain or change and has failed to uncover what it actually was that caused that knowledge change – was it the words, the presentation, the models, the presenter? If the authors have this information, it would be useful to include, or even provide some observations and reflections that might lead to further study. – Regarding the students feeling that the questionnaire was an exam, this is absolutely my experience too, particularly in areas where cultural differences between the students and the visitor are quite obvious. It would be really useful for the authors to reflect a bit on this, and ideally put it in the paper – might it be related to perceived or actual power, positionality, cultural norms etc?

**Response:** Thank you for these reflections. We exercised care when using the word significant, always clarifying what we meant (e.g., 95% level). This is also why we included the results from our statistical tests in the manuscript and referred the readers to the supplemental information for more details.  As for collecting feedback on desired video length, we have done this (please see section 4.2: Teacher feedback). In general, we observe that both the UK and Tajik teachers used the videos flexibly, skipping sections irrelevant to students' lives or shifting to lesson plans when the technology failed. As mentioned earlier in this documents, the video lessons are designed for a global audience. Therefore, teachers are encouraged to use/modify the videos according to the needs of their students and their local environment.

As for evaluating the causes of change in knowledge, this is discussed in section 4.1. We observed significant changes in students' knowledge of three concepts (earthquake location forecasting, Earth's interior and non-structural hazards). Each of these concepts and related changes are discussed in more detail in section 4.1.

Concerning the questionnaire being treated like an exam by Tajik students, we have added more reflection on this, as suggested by the reviewer, to section 4.3 (curriculum evaluation). This study shows that while using questionnaires with students may be a time/cost-efficient way of collecting information anonymously, if not designed carefully, the students may interpret the questions differently, as it was the case with some students in Tajikistan. This is especially important if the students are not experienced with participating in anonymous questionnaires where giving honest (as opposed to correct) information is crucial. Our recommendation, therefore, would be to use questionnaires as one of several methods for collecting and evaluating data. Amongst these methods, we recommend conducting face-to-face interviews with students when possible and/or arrange for group interviews. In this way, if a question is misinterpreted, the interviewer can rephrase it. In addition, recording the sessions may provide important insights into student-student and student-teacher interactions that enhance or hinder learning. We have added these recommendations to section 4.3.

Minor points to address:

Line 10 The opening line of the abstract is conjecture. Suggest a rewrite.
**Response:** Thank you for the comment. We agree, and have taken out this sentence.

Line 13 "coupled with activities carried out by local classroom teachers". Suggest a rewrite – the teachers aren't carrying out the activities, and what do you mean by local classroom? Isn't teachers enough?
**Response:** We have revised this sentence for more clarification: " […] coupled with activities carried out under the guidance of classroom teachers."

Line 21 I'd suggest replacing views with knowledge – you weren't evaluating their opinions. Although that would have been interesting. We have made this change.

Line 25 What makes a classroom culture suitable? Who says so? Suggest a rewrite.
**Response:** We have revised the text to clarify this: "[…] (e.g., levels of teachers' participation and classroom culture) […]"

Line 46 empowerment of communities
**Response:** Thanks for catching this typo.

Line 66 Be consistent with language – choose either teaching-duet or paired teaching
**Response:** We have replaced "teaching-duet" with "paired teaching" throughout the manuscript except in section 2.1 (line 80) where we make a reference to the MIT

BLOSSOMS (the original developer of this method which they sometimes refer to as teaching-duet).

Line 75 Typo on through
**Response:** We fixed this typo.

Line 87 How did you develop an iterative process? There did not seem to be any iteration in the UK example – did the video teacher actually talk to the teacher in-situ and tweak approaches? In the Tajik example the video teacher was the in-situ teacher, so again, no iteration.
**Response:** We agree with the reviewer, and have taken out this term. The final sentence reads, "The passing of teaching between the in-class and video-teachers is a type of blended learning referred to as the paired teaching."

Line 128 How do you know that the teacher continues to keep the students interested?
**Response:** Good point. We don't know. We have revised the sentence, "The teacher continues by stating that despite not being able to drill deep into the Earth's interior, […]".

Line 160 You collected data on the first and last days of video implementation but the testing periods were very different between the UK and the Tajik studies (50 and 5 days respectively). Can you reflect a bit on this, maybe in a later section?
**Response:** This is correct, and we have revised the manuscript to add our reflection to section 2.2.4 (school settings). The difference between the testing periods was due to different teaching schedules teachers had to follow. While the UK teachers were more flexible with their program and could spread out the video lessons in a 50-day period, the lead author of this manuscript had to follow a restricted schedule of five days for video testing in Tajikistan. It is possible that the differences in the testing periods between the two groups (UK vs Tajikistan) influenced the study results. For example, due to time constraints, some in-class activities had to be shortened or skipped in Tajik classrooms. Furthermore, technological issues in Tajik classrooms (e.g., system malfunction, missing cables, etc.) posed additional challenges to video testing by shortening video testing period.

Line 214 Section 2, not 3.
**Response:** We corrected this.

Line 293 How can you have a 20% increase in student understanding of the Earth's interior. Please clarify or omit this.
**Response:** Thanks for this comment. We have modified the manuscript to clarify this. The sentence now reads, "[…] indicate an increase of at least one score point in their understanding of […]".

Line 303 Might be a personal thing, but the word naïve feels a belittling – can the authors just say a basic conceptual framework?
**Response:** We agree and have changed the word naïve to a basic conceptual framework throughout the manuscript.

Lines 328-29 I like where the authors have put numbers of students in brackets, in really helps the reader. Consistency with this throughout the paper would be welcome.
**Response:** We have revised the text to be consistent throughout the manuscript.

Lines 397 I'd argue that the authors have not produced a series of DRR educational materials, only a few related to DRR and most were about hazard education.
**Response:** We changed this to "a series of earthquake education video lessons".

Line 409 constraints (typo)
**Response:** Thanks for catching this. We fixed it.

Line 411 Fix non-structural hazards? Could a more appropriate word be found?
**Response:** We are not aware of any other terminology that can be used to describe non-structural hazards. The term "falling hazards" is sometimes used to refer to non-structural hazards. However, this term is not entirely accurate as not all non-structural hazards are falling objects (e.g., tables not fixed to the floor can shift during earthquake shaking and cause injuries if they collide with people). In section 3.2.2., we have defined non-structural earthquake hazards as those "caused by the furnishings and non-structural elements of a building (e.g., suspended ceilings and windows)". We hope that this explanation is helpful to the readers.

Line 411 non-seismically active regions
**Response:** We made this change.

Line 420 "Nearly all Earth scientists agree that public outreach is important." How do the authors know this? Where's the evidence/reference? Otherwise just conjecture.
**Response:** We agree and have change this sentence to "The importance of public engagement activities is increasingly recognized by scientists, funding institutions, and policymakers", and provided additional references to support it: NSF, 2015; Rauws, 2015; European Union, 2002.

Line 460 I'd suggest changing the word ensure to support.
**Response:** We have made this change.

Line 460-466 These examples are ok, but are they really related to K-12 and higher education? How about researching some serious games such as Earth Girl and Hazagora?
**Response:** The reviewer is correct. These examples (i.e., Hazard Ready, Central Asia Geohazards Database, and GEM products) are not related to K-12 and higher education, but they are tools developed by scientists to enhance access to geohazards information. If incorporated into DRR educational materials, they have the potential to effectively serve as educational tools for increasing public knowledge on geohazards. The paired teaching approach, as discussed in this manuscript, is one way to turn such scientific tools into classroom video lessons. As for including examples of serious games such as Hazagora, we have added this information to section 1 (introduction). Thanks for bringing our attention to these and similar resources.

---

## Author Comment (AC2) · 5 Feb 2021

We thank the Anonymous Referee #1 for taking the time to review the manuscript (**gc-2020-43**: *Using paired teaching for earthquake education in schools*). We have updated the manuscript as suggested. The detailed response is shown below in **blue**.

1. The paper, in the middle, "goes into the weeds" on lots of little in-class details when the reader hungers for the "bottom line," i.e., Did it work? Yes or No? And why? Suggest that lots of these minutia details be relegated to an appendix, perhaps shortening the main text by 50% or so. The main text can summarize "the weeds."

**Response:** We thank the reviewer for this observation. However, we are not sure what exactly has been identified as "minutia details" that should be moved into an appendix. The middle of the paper (referred by the reviewer) is the result section. We shortened the text, but could not remove a large section from this part of the manuscript as it describes the evaluation data and plots. We have also moved section 2.1.1 to the appendix. This section provides an example of a paired teaching video which is explained in 8 paragraphs.

2. ". . .the local language (Tajik) was used in teaching and in all written and media materials." Does this mean that the paired teaching videos were available in Tajik as well as in English? The manuscript is unclear on this. It states that the live discussions the of students and teachers were in Tajik, but is unclear on the videos. If the students in Dushanbe were shown English-spoken videos, then the lack of before-and-after knowledge improvement is most understandable.

**Response**: Yes, the paired teaching videos that were tested in Tajik classrooms in were dubbed in Tajik (the local language). We have modified the manuscript text to make this point clear.

3. The key message is the "underwhelming" amount of new learning, especially in Dushanbe. This reviewer was unable to discern how much of this is due to language (were the videos in English?) and how much to culture. Interactive classrooms are quite novel in most countries – where the tradition is that the teacher is undisputed leader and the students dutifully follow. Perhaps it would have been better to start the students on paired-learning materials on more traditional STEM subjects like math or basic science. But this is easier said than done due to lack of video materials in the local language.

**Response:** Thank you for making this point. We do not think that the language was a barrier to learning. As mentioned under our response to comment #2, all instructions and discussions as well as materials used for teaching including the videos were made available in the local language. As for other hindering factors, we agree with the reviewer that interactive classrooms are novel in most countries where many teachers are unfamiliar or inexperienced with collaborative learning methods. This certainly was the case in Tajikistan. Therefore, the textbook-based classroom culture is most likely an important hindering factor in this study. We have discussed this in more detail in section 4.3 (classroom culture).

As for using more traditional STEM subjects to introduce paired teaching, this can be a possibility as long as the content is relevant and related to those subjects. Based on the previous experiences and input we received from the school administrators and teachers, geography classes were a good choice for testing our geoscience videos because of the content overlap. The videos covered topics that were also covered (though not in depth) in geography textbooks (e.g., Earth's interior, plate motions, and natural hazards).

4. The failure of teachers in Dushanbe to serve as in-classroom teachers in the paired-teaching mode may be cultural and/or due to lack of training in this new pedagogical model. Please discuss.

**Response:** We agree with the reviewer, and have added a paragraph to the manuscript to discuss this point. We provide a summary here: the paired teaching video lessons are designed to be a complete resource. They include video segments, teacher's guides, downloadable handouts and lists of other resources relevant to the topic. Therefore, no teacher training should be needed in order for teachers to use these videos. However, teachers are encouraged to view the videos and familiarize themselves with the content before using them in their classrooms. This study, however, reveals that these videos may not be seen as a complete resource by some teachers. While the UK teachers tested the videos with minimal input from video creators, teachers in Tajikistan asked to observe classroom testing of the videos. This request was made despite the fact that the teachers were offered (i) training to deliver the videos, and/or (ii) the option to co-teach the video lessons with experienced instructors. The textbook-based classroom culture (typical of schools in Tajikistan) may explain why Tajik teachers did not want actively engage in video testing. The study, therefore, shows that the paired teaching pedagogy is not a "one size fits all" teaching approach, and depends on the classroom culture and teacher's comfort operating within it.

5. The recently renovated school in Dushanbe: Was it designed with the latest protections for earthquakes, with their destructive threats to life and limb? Answering that question would have been a nice addition to class discussions, especially if the answer was very positive – that is, substantial structural improvements and student training improvements.

**Response:** The school in Dushanbe was not renovated. It is a brand new school built in the place of the old school. We have changed the language in the manuscript to correct this. The new school has been constructed according to the existing seismic building codes of Tajikistan. Since schools are considered to be critical infrastructures (like hospitals) in Tajikistan, they are designed to withstand earthquakes with intensity degree of IX on Medvedev-Sponheuer-Karnik (MSK-64) intensity scale. The IX intensity value describes earthquake events that can be destructive, causing substandard structure collapse and substantial damage to well-constructed structures. Therefore, the school has been built with the latest protections for earthquakes.

The structural integrity of the school building was not discussed with the classroom during video testing as it was not the focus of the three selected videos. However, as part of our non-structural hazards video lesson, students were given the opportunity to carry out a rapid visual screening of their classrooms to identify and evaluate non-structural hazards. These are non-structural components in buildings such as furnishing, equipment, electrical and mechanical fixture and architectural features. For each non-structural hazard students identified, they offered possible mitigation strategies (see section 2.2.3, lines 196-200). The non-structural hazard video lesson uses a place-based approach, which promotes the kind of discussion the reviewer is recommending.

---

## Author Response (AR2)

We thank Katharine Welsh for her comments on the manuscript (**gc-2020-43**: *Using paired teaching for earthquake education in schools*). We have updated the manuscript with the information requested. Our response is also shown below in **blue**.

Ln57 - please amend to include the citation rather than stating "According to the above report..."

We agree and have modified this sentence to state: "While some DRR content is easily woven into specific school subjects such as geography or natural sciences, a textbook-driven approach hinders the achievement of skills, attitudinal and action learning outcomes required for effective DRR learning." See lines 55-57.

Ln 185 - please could you detail the ethical challenges and mitigations here (within the paper) as you have done (very nicely) in the author responses

We added this information to section 2.2 (lines 111-120). The new information is also shown below.

"Ethical approval for this study was sought and received from the participating schools and institutional partners that coordinated this effort in 2018-2019. The ethical procedures were designed to adhere to current standards of assent and consent regarding in-school research and to provide participants with anonymity. As a low-risk, school-based study focused on learning outcomes from regular teaching activities, the consent of the school-based stakeholders, i.e. the principals and teachers, were considered sufficient to proceed with the pre- and post-tests. All participation was voluntary, and students were given the opportunity to assent or refuse participation at both the pre- and post-test points in time. The pre- and post-tests were anonymous with students creating their own codenames, which were known only to them, that were used to match the pre- and post-tests for analysis purposes. No sensitive or identifying information was collected, and the anonymous data are stored in a secure location within the European Union that is password protected, in accordance with GDPR regulations. The data will be destroyed upon completion of this research project."

Ln 461 - is there evidence in the literature (i.e. in other countries) that you could use to support your suggestion here?

Good point. There is evidence for supporting our suggestions. We have modified section 4.3 (lines 385 – 404) to include references to previous studies that have tested paired teaching in other countries including China, Japan and Malaysia. The modified text is also shown below.

"The low level of engagement by local teachers in Tajikistan in serving as in-class teachers in the paired-teaching approach may be due to their unfamiliarity and discomfort with collaborative learning methods and the use of video technology. Since the paired teaching video lessons were designed to be a complete resource (i.e., containing video segments, teacher's guides, downloadable handouts and lists of other resources relevant to the topic), no teacher training was provided for using these videos. However, teachers were encouraged to view the videos and familiarize themselves with the content before using them in their classrooms. This study, however, reveals that these videos may not be seen as a complete resource by some teachers. While the UK teachers tested the videos with minimal input from video creators, teachers in Tajikistan asked to observe classroom testing of the videos. This request was made despite the fact that the teachers were offered training to deliver the videos, and/or the option to co-teach the video lessons with experienced instructors. Similar to teachers in Tajikistan, teachers in China, Japan and Malaysia, where

rote learning dominates classroom culture, experienced difficulties with paired teaching (Larson and Murray, 2017). Therefore, the textbook-based classroom culture may partly explain why Tajik teachers did not want to actively engage in video testing. In addition, teachers' low level of technology acceptance and readiness for teaching and learning has been shown to hinder their engagement with technology-based pedagogical approaches (Shukor et al., 2018). Our study, therefore, shows that the paired teaching pedagogy is not a "one size fits all" teaching approach, and depends on the classroom culture and teacher's comfort operating within it. Taken together, when developing curricular material, teachers' and students' involvement are key to ensuring an appropriate selection of content and pedagogical approaches. This can be achieved through informal classroom observation and discussions of goals and pedagogical expectations with classroom teachers and students as well as providing ongoing, high-quality pedagogical training that support teachers with adopting a more student-centered and collaborative teaching style for their classrooms."

***END OF COMMENTS***